# Influenza Vaccination of Nurses and Other Health Care Workers in Different Occupational Settings: A Classic and AI Mixed Approach for Time-to-Event Data

**DOI:** 10.3390/nursrep15030087

**Published:** 2025-03-03

**Authors:** Matteo Ratti, Riccardo Rescinito, Domenico Gigante, Alberto Lontano, Massimiliano Panella

**Affiliations:** 1Department of Translational Medicine (DiMeT), Università del Piemonte Orientale, 28100 Novara, Italy; 10033325@studenti.uniupo.it (R.R.); domenico.gigante@uniupo.it (D.G.); massimiliano.panella@uniupo.it (M.P.); 2Azienda Ospedaliero Universitaria Maggiore della Carità, 28100 Novara, Italy; alberto.lontano01@icatt.it; 3Occupational Safety and Prevention Service (SPreSAL), ASL VC (Local Health Authority), 13100 Vercelli, Italy; 4Section of Hygiene, Department of Life Sciences and Public Health, Università Cattolica del Sacro Cuore, 00168 Rome, Italy; 5Azienda Ospedaliero Universitaria SS. Antonio e Biagio e Cesare Arrigo, 15121 Alessandria, Italy

**Keywords:** seasonalinfluenza, vaccination, artificial intelligence, health care workers, occupational safety

## Abstract

**Background:** Seasonal influenza currently remains a major public health concern for the community and, in particular, the health care worker (HCW). According to the World Health Organization, HCWs are among the high-risk categories for which vaccination is recommended, due to the derived absenteeism, productivity loss, and high probability of transmitting the disease to vulnerable individuals or patients. Therefore, an HCW vaccination policy should be adopted by every health care provider. There is growing evidence that a time effect of the vaccination event is probable, which may influence vaccine effectiveness. We designed and conducted an observational study to investigate the time to anti-influenza vaccination event of different categories of HCWs belonging to different occupational settings in a tertiary hospital during three seasons in order to retrieve some insight about HCW prioritization when designing vaccination campaigns. **Materials and Methods:** We retrospectively analyzed the results of two HCW anti-influenza vaccination campaigns (2022 and 2023) to assess any difference regarding job typology and unit typology (critical care, surgical, medical, service). We first fitted a classic Cox proportional hazard model and then an AI random forest model to assess variable importance. We used R, RStudio, and the survex package. **Results:** Overall, other HCWs reported a lower vaccination rate compared to nurses (HR 0.77; 95%CI 0.62–0.97), and service unit personnel appeared to more likely be vaccinated (HR 1.42; 95%CI 1.01–1.99) compared to those belonging to the critical care units. As expected, older workers tended to be vaccinated more frequently (HR 1.70 for the (46, 65] category compared to the younger one; 95%CI 1.39–2.09). The variable importance analysis showed consistent superiority of the ward typology and age category variables with respect to time. During the entire timeline, the ward typology appeared to be more important than the HCW typology. **Conclusions:** Our results suggest a prioritization policy based firstly on the unit typology followed by the job typology for HCW anti-influenza campaigns.

## 1. Introduction

Seasonal influenza remains a major public health concern for the community and, in particular, the health care worker (HCW). The European Centre for Disease Prevention and Control (ECDC) estimates that, in the European Union (EU) every year, this pathogen is responsible for up to 50 million symptomatic infections, which, in turn, lead to 15,000–70,000 deaths [1]. According to the World Health Organization (WHO), HCWs are among the high-risk categories for which vaccination is recommended, due to the infection-related absenteeism, productivity loss, and high probability of transmitting the disease to vulnerable individuals or patients [2]. In fact, it has been estimated that HCWs carry a 10-fold increased risk of influenza [3]. Another multicenter prospective cohort study conducted in five French university hospitals calculated a cumulative influenza incidence of 22.3% among HCWs [4], compared to a recent estimate for the general population of 3–11% [5]. Moreover, the role of the HCW in spreading influenza appears to be confirmed by the literature. A study with a seven-season timespan regarding 62,343 hospitalized patients reported that, while influenza vaccination coverage among health care workers decreased from 13.2% to 3.1%, the frequency of nosocomial Influenza-Like Illness (ILI) in hospitalized patients increased from 1.1‰to 5.7‰ [6]. Another study found that a ≥35% vaccinated proportion of HCWs appeared to protect against hospital-acquired influenza among short-stay unit patients (OR 0.07) [7]. Another recent systematic review and meta-analysis study by Li and colleagues yielded a combined RR regarding the incidence of laboratory-confirmed influenza of 0.36 (95% CI: 0.25 to 0.54) [8]. Finally, even mathematical modeling of influenza spreading in confined facilities yielded promising results, showing a positive effect of HCW vaccination against transmissions of the disease [9,10].

Due to the fact that the timing of the seasonal influenza peak varies substantially among years [11], many efforts have been made in order to correctly forecast influenza activity, often with the help of complex mathematical models parametrized with surveillance data [12]. For instance, the efforts of the US Centers of Disease Control and Prevention (CDC) in forecasting started in 2013 with a competition open to both public and private participants [13] and are now structural, highlighting the fact that, despite their intrinsic imperfection, the predictions provide some useful information for designing vaccination campaigns. Given this uncertainty, though, some authors began to investigate the timing of vaccination. A recent study about waning protection explored the consequences of a change in timing for older adults [14]. Even though the authors concluded that a too early vaccination may increase the hospitalization rate, they did not find enough evidence to support a policy change yet. Another proof of a time effect of the vaccination event regarding HCWs is provided by the study of Murti et al. [15]. HCWs who underwent a vaccination event before December (’early’) were less likely to take sick time (OR 0.874) and reported less sick time (RR 0.907) compared to HCWs who did not report vaccination. Moreover, HCWs who reported being vaccinated between 1 December and 31 March of the following year (’late’) had similar sick rates to HCWs who were not vaccinated. Another study by Sadeeh et al. [16] found a statistically significant association between earlier influenza vaccination among HCWs and a reduction in ILI, leaves of absence, and days of lost work. In summary, even if the direction of the effect is still debated, a time effect of the vaccination event on ILI and influenza transmission rates seems more than probable.

The risk of contracting a respiratory disease differs among health care workers providing care in different hospital departments. A recent meta-analysis by Tian et al. [17] found an increased risk for frontline workers compared to non-frontline workers (OR 1.66), though it did not identify significant differences in respiratory infection risks among the professional categories analyzed. Conversely, a study by Torén et al. [18] highlighted an increased risk of respiratory infections among health care professionals compared to unexposed individuals, distinguishing by professional macro-profile. Specifically, the risk was higher for physicians (OR 3.21), followed by nurses (OR 1.72) and other health care workers (OR 1.75). Moreover, the risk of infection is higher among workers performing aerosol-generating procedures (OR 2.5), such as non-invasive positive-pressure ventilation, cardiopulmonary resuscitation, manual ventilation, bronchoscopy, and suctioning [19]. These procedures are implicated in the transmission of various pathogens responsible for respiratory infections, including influenza [20], and are only performed in particular wards of hospitals, mainly surgical or critical care units.

The WHO currently recommends a policy for the vaccination of high-risk individuals, including HCWs. An international survey reported that 53 Member States had already implemented an HCW national policy, 10 had declared plans to introduce it within 5 years, 20 had subnational/institutional policies, and 22 had no policy for vaccinating HCWs [21]. In Italy, the national vaccination plan 2023–2025 not only recommends that health care facilities have a vaccination policy for HCWs, but also suggests risk assessment to prioritize access [22]. Therefore, any difference in the behavior of HCWs towards the vaccination event should be studied in depth to fully understand the risk to different individuals.

In summary, to design an effective HCW vaccination campaign, it seems important to investigate not only the overall coverage rate, but also the time to vaccination event, discriminating by professional category and unit typology. To our knowledge, no studies in the literature have evaluated the timeliness of patient acceptance of anti-influenza vaccination yet. We therefore designed and conducted a retrospective study with the aims of assessing any difference among HCW typologies with respect to the time to vaccination event; investigating any difference among HCWs belonging to units of different care intensity or nature; and identifying the most important variables which lead to the vaccination event over time. In particular, we were interested in assessing if the working setting was more or less important than the categorization of the working tasks, in order to identify a possible prioritization rule for the design of vaccination campaigns.

## 2. Materials and Methods

### 2.1. Study Design and Aim

We conducted a retrospective observational study of recent local vaccination campaigns against seasonal influenza performed at a tertiary-level teaching hospital. The study design was an observational historical cohort study. The aims of the study were (1) assessing any difference between nurses and other health care professionals with respect to the time to vaccination event, (2) assessing any difference among HCWs belonging to units of different care intensity or nature, (3) identifying the most important variables which lead to the vaccination event over time. In particular, we were interested in assessing if the working setting (ward typology) was more or less important than the categorization of the working tasks (HCW typology).

### 2.2. Study Setting and Timeline

The study was conducted at the University of Eastern Piedmont “AOU Maggiore della Carità” Teaching Hospital, Novara, Piedmont, Italy. The hospital had 657 beds and employed 3115 workers as of December 2024. We anonymously collected the data regarding the influenza vaccination of the nurses and other health care workers for the winter seasons of 2022 and 2023. The start date of the campaign was set according to the one for the general-population anti-influenza campaign defined by Regione Piemonte, the regional health authority responsible for the delivery of the vaccine to general practitioners and other health care local providers such as hospitals, long-term care facilities, and public health districts. The 2022 date was set as 24 October while the 2023 starting date was 16 October. The general-population campaigns lasted until February of the following year, but the health care workers’ low demand compared to the organizational resources caused the end of the local campaigns on 15 December. The hospital faced no delay in starting the campaign or in receiving the vaccine. The vaccine used was a quadrivalent inactivated viral antigen (Influvac S Tetra 0.5 mL—Viatris Healthcare Ltd., Canonsburg, PA, United States).

### 2.3. Inclusion and Exclusion Criteria

We included in the study the health care workers of the hospital that were actually in service in the month (at least one day of presence) before the official start of the campaign. We excluded physicians and resident physicians due to incomplete data on these professionals.

### 2.4. Data Extraction

After the campaign, a researcher obtained the oral informed consent of the workers and anonymously collected the following data from the hospital information system about the nurses/HCWs: date of vaccination event (with the calculated time to vaccination), gender, age, ward typology. We categorized the age into two categories depending on the median value of the whole sample. We classified both the health care workers and the unit they were working in as follows. We used RedCap to compile the database.

#### 2.4.1. Classification of Health Care Workers

We classified the HCWs into two groups: “nurses” and “other HCWs”. The former category included both general and specialized (pediatric) nurses, while the latter included the following professionals: physiotherapists, occupational therapists, laboratory technicians, radiology technicians, health care operators, prevention technicians, biologists, psychologists, pharmacists.

#### 2.4.2. Classification of the Hospital Units

The units of the hospital were grouped according to the Ministry of Health classification of medical specializations: the ICU and other high-intensity areas (such as the emergency department) were marked as “critical care units”; all the medical disciplines such as internal medicine, pneumology, neurology, and onco-hematology were classified as “medical units”; all the surgery-related disciplines (abdominal surgery, plastic surgery, thoracic surgery, etc.) were grouped together under the “surgical unit” category; finally, radiology, rehabilitation, and the laboratories were labeled as “service units” [23].

### 2.5. Data Analysis—Classic Statistics and Cox Model

We described the continuous variables of our sample by means (or medians) and percentages according to the nature of the single covariates. We compared the participants’ characteristics with *t*-tests or chi-square tests as appropriate. In the case of not normally distributed variables, we employed the Kruskal–Wallis test. We calculated the Kaplan–Meier curves and the relative product limit of the vaccination event by HCW typology and unit typology. We finally fitted a Cox proportional hazard univariable and multi-variable model. We employed all the variables and checked the proportional hazard assumption of the model with the scaled Schoenfeld residuals. All the data were analyzed with R ver. 4.4.0 [24] and RStudio ver.2023.09.1 Build 494 [25], along with the packages tidyverse, survival, and survminer.

### 2.6. Data Analysis—AI Model for Variable Importance Assessment

To address the time-to-vaccination variable’s importance, we fitted a random forest AI model with the whole dataset and compared the performance to the classic Cox model with the following metrics: C-index, Integrated C/D AUC, and Integrated Brier Score. The C-index, or concordance index, is one of the most commonly used metrics for the overall evaluation of survival prognostic models [26]. It is a common interpretation, similar to the AUC of ROC curves, to consider 1 as perfect concordance between the proportion of events and the relative times and 0 as the perfect anti-concordance. The mid value of 0.5 therefore denotes a random classifier. The C/D (cumulative/dynamic) AUC, one of the other common metrics, is the probability that two randomly chosen participants, one having experienced the event before time t and the other one after t, are correctly classified. Again, the integration of the metric ranges from 0 to 1, indicating poor and perfect performance, respectively [27]. The Brier score, finally, is used to evaluate the accuracy of a predicted survival function at a given time t and it consists of the mean squared error between the observed status and the predicted probability. The integrated Brier score is a number between 0 and 1 as well, but 0 is the best value [28]. The requirement for further analysis of the variable importance was due to the non-inferiority of the random forest model fit with respect to the Cox model as measured by these metrics. Then, we planned to explore the variable importance with the Brier score loss after covariate permutations and graphically analyze the trend of importance with respect to time. All the analyses were conducted with the survex package ver. 1.2.0 [29] and, for the AI model, the ranger package ver. 0.17.0 [30]. This package contains a fast implementation of a survival random forest algorithm.

### 2.7. Ethical Considerations

We conducted the study according to the principle of the Declaration of Helsinki and the EU data protection regulation GDPR 679/2016. Given the retrospective observational design, according to national law (GU no. 76 of 31 March 2008), ethical committee approval was not necessary. All the data collected were anonymous, therefore it was not possible, even for the researchers, to reassign a single observation to an individual. However, prior to the beginning of the study, the local ethical committee was consulted, and, before the data collection phase, a researcher obtained the workers’ oral informed consent.

## 3. Results

Our whole sample of nurses and other health care workers consisted of 2982 units, 1444 (48.4%) present during the 2022 campaign and 1538 (51.6%) during the 2023 campaign. The whole sample showed a predominance of the female gender (81.4%) and nurses (72.6%). No differences were found between the two seasons for HCW typology (73.5% of nurses in 2022, 71.7% in 2023; *p* value: 0.3), age category (43.6% of young workers in 2022 vs. 45.8% in 2023; *p* value: 0.239), or ward typology (critical care unit 24.6% in 2022 vs. 24.1% in 2023; *p* value: 0.929). The full characteristics are reported in Table 1.

The overall vaccination events out of the grand total (multi-season) denominator resulted in a proportion of 14.9% (443/2982) with a median time to vaccination of 25 days (IQR: 18–37). The 2022 campaign had 224 events out of 1444 workers (15.5%, median time to vaccination of 24 days, and IQR 17–32). During the 2023 year, we observed 219 events among 1538 individuals (14.2%; median time to vaccination 29 days; and IQR 23–40). As shown in Table 2, there were no significant differences in the chi-square test between the distributions of the HCW categories for the unvaccinated and vaccinated individuals, or for the gender proportions. The vaccinated group showed a tendency towards the older age category and a little significant ward typology difference.

The proportions of vaccinated individuals among workers and ward typologies showed a difference in distribution. For the critical care unit, 13.3% of nurses and 10.3% of other HCWs underwent influenza vaccination, whereas, in the surgical unit, it was 15.4% and 9.3%, respectively. The medical unit reported an uptake of 17.2% for nurses and 12.4% for other HCWs. Finally, in the service unit, 13.3% of nurses and 22.4% of other HCWs underwent an anti-influenza vaccination. The time to vaccination event ranged from 24 (nurses of critical care units and surgical units) to 30 days (nurses of service unit) from the beginning of the campaign. The full summary statistics about the vaccination events are reported in Table 3.

The product limit Kaplan–Meier estimator, along with the specific survival tables, is reported in Appendix A. The relative event-free probability curves are reported in Figure 1, and differ significantly (log-rank test *p* value = 0.0013).

The results of the univariable Cox model fit are reported in Appendix A. The significant variables were age category (HR 1.7, 95%CI 1.4–2.1) and ward typology (service unit HR 1.5 vs. critical care unit, 95%CI 1.09–2.1). The multi-variable Cox proportional hazard (PH) model fit yielded significant differences for the other HCW group (HR 0.77; *p* value = 0.024), the service unit compared to the critical care unit (HR 1.42; *p* value = 0.043), and the older age category (HR 1.7; *p* value < 0.001). No significant difference was found for gender (male HR 0.83; *p* value = 0.166). The forest plot of the multi-variable Cox model is shown in Figure 2. The scaled Schoenfield residuals examination resulted in a non-significant association with time (global test *p* = 0.42; HCW typology *p* = 0.45; ward typology *p* = 0.16; gender *p* = 0.49; age category *p* = 0.84), therefore validating the assumption of proportional hazards. The relative plot is reported in Appendix A.

The random forest model fit yielded an overall out-of-bag (OOB) error of 0.43. All the other parameters of the model are reported in Appendix A. The relative performance metric values with respect to the Cox model are reported in Table 4. The random forest model exhibited better performance for all three metrics, thereby achieving the non-inferiority requirement for further analysis.

The variable importance plot (Figure 3) shows the consistent superiority of the age category and ward typology variables with respect to time. During the entire timeline of the campaign, the ward typology appeared to be more important than the HCW typology.

## 4. Discussion

We designed and conducted a retrospective study concerning the campaign for the anti-influenza vaccination of HCWs in a tertiary hospital. Our result regarding the vaccination event was an overall coverage ranging from 9.3% to 22.4% across the ward and HCW typologies. We conclude that the nurse/HCW vaccination rate, although it experienced moderate variability, remained suboptimal, given the recommendation of the WHO [2] and CDC [31] that all HCWs should be vaccinated against seasonal influenza. The literature concerning this aspect confirms our finding. A study in a southern Italian hospital found that, in the 2018/19 season, the anti-influenza coverage was only 20.4%, although it was higher than the 14.2% of the preceding season. The highest vaccination rate was found among physicians (33.4%), followed by other HCWs (23.8%) and nurses (7.2%) [32]. Another multi-year (2015–2018) national survey (ministerial surveillance) found a vaccination proportion of 8.5% for non-medical health professionals, with a slightly increased value for older age categories [33]. Although vaccination coverage experienced a moderate increase during the COVID-19 years [34,35], our findings corroborate a return to the pre-pandemic rates already described in the literature [36]. Interestingly, the coverage values that we found are substantially lower than those of other countries. A study from the US calculated 36.6% for nurses and 48.9% for other HCWs [37]; another German survey found 39.3% for HCWs as a whole [38]; and, finally, a Swiss research group found 40.2% (78.1% of physicians, 47.3% of pharmacists, 29.1% of nurses, 24.3% of technicians) [39]. Notably, in many of these countries, HCW vaccination is mandatory either for legal or insurance reasons. This highlights the question of a mandatory vaccination policy for HCWs, an intervention which has proved its effectiveness despite still being debated in the literature, mainly for ethical reasons [40,41,42].

In fact, a mandatory vaccination policy has been proved to be the most effective intervention to decrease vaccination hesitancy (calculated RR of being unvaccinated of 0.18, 95%CI 0.08–0.45), followed by “soft mandates” such as declination statements and, finally, increased awareness and access, as shown by a systematic review and meta-regression study [43]. These findings are corroborated by other evidence. For instance, a Canadian 10-year mandatory policy study found a sustained increase in immunization rates among HCWs [44]. The scientific literature about the topic, moreover, seems to suggest a shift also in occupational groups, as we did. A Swiss study found different barriers to vaccination among four HCW categories. The authors concluded that such differences must be taken into account when designing a vaccination campaign [45].

The log-rank test and the multi-variable Cox model suggest that workers from the units marked as “service” are prone to be vaccinated more frequently when compared to the ones belonging to the “critical care” category. Some studies found that the coverage rate was higher in medical units, followed by service units and, finally, surgery units [32,46]. Another study found a higher proportion of vaccinated HCWs in service units compared to the surgical unit [47] but no statistically significant coefficient in the logistic regression analysis. Because of the relatively scarce contact of the “service units” workers compared to the other ones [48,49], our results interestingly show that the categories which carry less risk of contracting or spreading the disease tend to be vaccinated more frequently. Given this counterintuitive finding, we reasonably think that, when designing a vaccination campaign, the prioritization policy should take into account the contact patterns of HCWs to improve its effectiveness.

The random forest model fit had a non-inferior performance when compared to the classic Cox model. While the latter is semi-parametric, the AI algorithm is a purely non-parametric technique, resulting in an entirely data-driven model. Given this adequate performance, we used this model fit to assess the variable importance. This analysis was already recently employed successfully to predict survival among 1159 heart failure patients. The authors of the study concluded that the explainable machine learning model is superior to the classic Cox model [50]. Our results confirm this finding and extend the range of use to the time to vaccination event. Even if, in our opinion, this kind of analysis will not overtake the classic statistic in confirmatory analysis, it will probably give some useful insights by means of exploratory ones.

Based on the variable importance analysis, we concluded that the ward typology variable contribution is superior to whether the worker belongs to the “nurse” or the “other HCW” categories during the timeline of the campaign. The age category seems to clearly dominate the predictions from the start of the campaign. This, coupled with the significance of the same variable in the classic Cox model and the result of the vaccination coverage, may suggest ward prioritization rather than job prioritization in the design of vaccination campaigns. Moreover, a prioritization by age seems to be advisable. No such studies were found in the literature to compare with this finding. Further research, with a more focused study design, is needed to validate our results. However, even if we can conclude that the time to vaccination event of the HCWs is determined more by ward typology, our study was not designed to assess an order of prioritization, which should take into account also the effectiveness of the vaccination campaign with regards to the main outcomes of vaccination (ILI or absenteeism rate).

### Strengths and Limitations

Our study has several strengths and limitations. To our knowledge, this is the first attempt to model the time to the vaccination event. This aspect may be important, as we found enough evidence in the literature that suggests differences in vaccine effectiveness depending on when the vaccination event occurred. Second, given a sufficient large sample size, the novel AI/ML techniques can provide meaningful results without relying on the classic statistic assumption. Few studies in the literature analyzed these data with such techniques. Our study employs both classic semi-parametric statistics and innovative data-driven modeling, comparing the performance of the two. However, the main limitation of our study is that we only considered the timeline of the vaccination campaign and what happened inside it. Therefore, we cannot exclude that a significant proportion of the HCWs were vaccinated outside the hospital (e.g., at their general practitioner’s office) or after the end of the campaign. The HCWs were not obliged to notify the hospital about their vaccination status. Moreover, we included in the denominator only the HCWs and nurses who had at least one day of presence in the month preceding the start of the campaign. Therefore, any worker on sick or holiday leave that may have returned to work during the campaign was excluded from the count. Given these factors, caution should be used in interpreting our results as an absolute measure of the vaccine coverage of the considered seasons. Finally, without an effectiveness measure of the campaigns, no order of prioritization can be determined yet.

## 5. Conclusions

The WHO suggests that every health care facility should have a policy for the anti-influenza vaccination of its HCWs and provides a specific implementation guideline [51]. In addition, the same institution recommends a risk stratification for HCWs in the design of such programs. Our study demonstrated that the overall coverage is still at a suboptimal level despite the increase in vaccination events during the pandemic years. Therefore, public health policy institutions must adopt adequate countermeasures to decrease the hesitancy given the evidence that an increase in coverage yields favorable health outcomes both for the patients and for the worker him/herself [8]. Our results give some insight into how to categorize and prioritize workers during influenza vaccination campaigns. Even if an outcome measure of effectiveness must be taken into account to fully evaluate a vaccination campaign, our results suggest a prioritization policy based firstly on the unit typology followed by the job typology.

## Figures and Tables

**Figure 1 nursrep-15-00087-f001:**
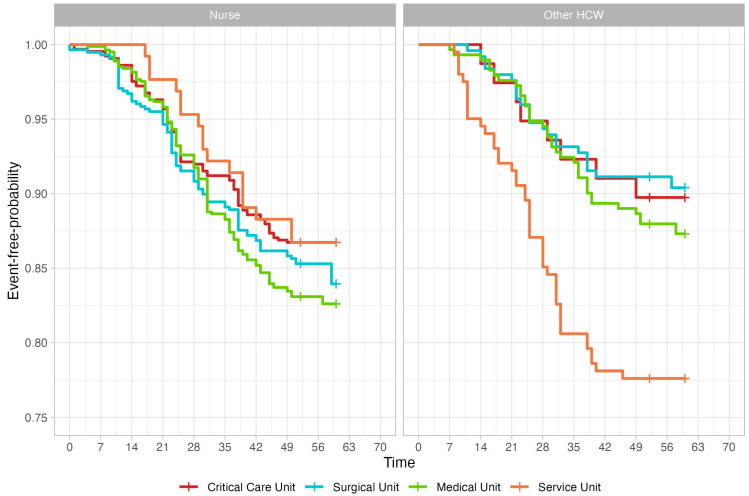
Kaplan–Meier curves of different unit typologies (colors) and worker typologies across the anti-influenza vaccination campaigns.

**Figure 2 nursrep-15-00087-f002:**
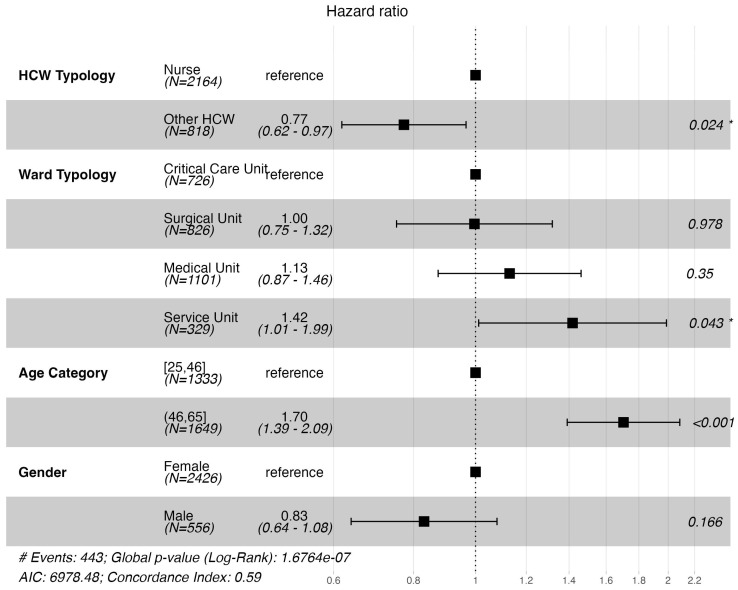
Forest plot of the results of the multi-variable Cox PH model. The * indicates significant *p* values between <0.001 and 0.05.

**Figure 3 nursrep-15-00087-f003:**
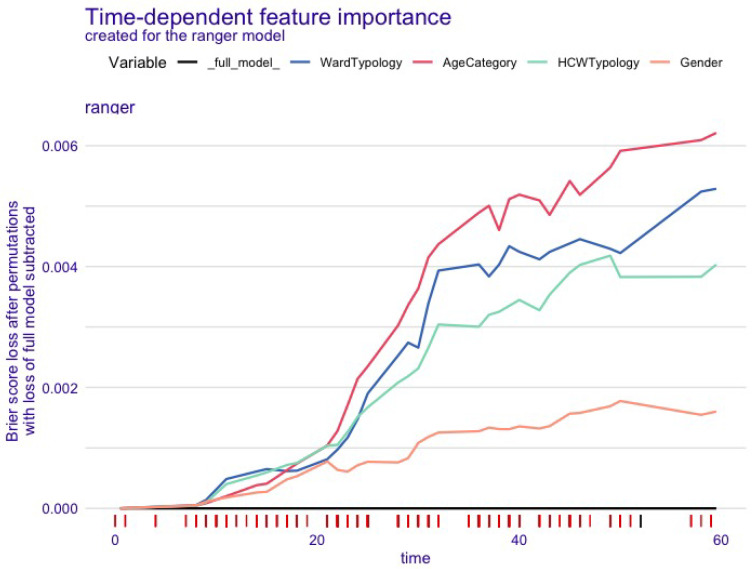
Variable importance plot of the random forest model fit.

**Table 1 nursrep-15-00087-t001:** Main baseline characteristics of the health care worker participants during the two campaigns.

	Levels	Overall	2022	2023	*p*
n		2982	1444	1538	
HCW Typology n (%)	Nurse	2164 (72.6)	1061 (73.5)	1103 (71.7)	0.3
	Other HCW	818 (27.4)	383 (26.5)	435 (28.3)	
Gender n (%)	Female	2426 (81.4)	1175 (81.4)	1251 (81.3)	1
	Male	556 (18.6)	269 (18.6)	287 (18.7)	
Age Category n (%)	[25, 46]	1333 (44.7)	629 (43.6)	704 (45.8)	0.239
	(46, 65]	1649 (55.3)	815 (56.4)	834 (54.2)	
Ward Typology n (%)	Critical Care Unit	726 (24.3)	355 (24.6)	371 (24.1)	0.929
	Surgical Unit	826 (27.7)	396 (27.4)	430 (28.0)	
	Medical Unit	1101 (36.9)	538 (37.3)	563 (36.6)	
	Service Unit	329 (11.0)	155 (10.7)	174 (11.3)	

**Table 2 nursrep-15-00087-t002:** Main characteristics of the unvaccinated vs. vaccinated HCWs during the two campaigns.

		Not Vaccinated	Vaccinated	*p*
n		2539	443	
HCW Typology n (%)	Nurse	1833 (72.2)	331 (74.7)	0.298
	Other HCW	706 (27.8)	112 (25.3)	
Gender n (%)	Female	2051 (80.8)	375 (84.7)	0.062
	Male	488 (19.2)	68 (15.3)	
Age Category n (%)	[25, 46]	1188 (46.8)	145 (32.7)	<0.001
	(46,65]	1351 (53.2)	298 (67.3)	
Ward Typology n (%)	Critical Care Unit	632 (24.9)	94 (21.2)	0.041
	Surgical Unit	714 (28.1)	112 (25.3)	
	Medical Unit	926 (36.5)	175 (39.5)	
	Service Unit	267 (10.5)	62 (14.0)	

**Table 3 nursrep-15-00087-t003:** Influenza vaccination status of nurses and other health care workers.

		Critical Care Unit		Surgical Unit		Medical Unit		Service Unit		
	Levels	Nurses	Other HCWs	Nurses	Other HCWs	Nurses	Other HCWs	Nurses	Other HCWs	*p*
n		648	78	578	248	810	291	128	201	
Vaccinated n (%)	yes	86 (13.3)	8 (10.3)	89 (15.4)	23 (9.3)	139 (17.2)	36 (12.4)	17 (13.3)	45 (22.4)	0.002
	no	562 (86.7)	70 (89.7)	489 (84.6)	225 (90.7)	671 (82.8)	255 (87.6)	111 (86.7)	156 (77.6)	
Time to vaccination (median [IQR])		24.00 [18.00, 38.00]	26.00 [20.75, 34.00]	24.00 [15.00, 38.00]	25.00 [21.50, 33.50]	29.00 [22.00, 37.00]	29.50 [23.00, 38.00]	30.00 [25.00, 39.00]	25.00 [15.00, 31.00]	–

**Table 4 nursrep-15-00087-t004:** Performance metric of the two model fits: Cox proportional hazard and random forest.

Model	C-Index ^1^	Integrated C/D AUC ^1^	Integrated Brier Score ^2^
Cox PH	0.5550325	0.5452379	0.1193034
Random F	0.5819838	0.6399568	0.1176841

^1^ Values close to 1 denote better performance. ^2^ Values close to 1 denote worse performance.

## Data Availability

The dataset is available upon request. The dataset contains only anonymous data; therefore, no worker ID or any reference to a specific individual is present. Reconnecting a single observation to the relative individual is impossible even for the researcher.

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
