# Peer review of "Influenza Vaccination of Nurses and Other Health Care Workers in Different Occupational Settings: A Classic and AI Mixed Approach for Time-to-Event Data"

_nursrep, 2025, doi:10.3390/nursrep15030087_

Round 1
Reviewer 1 Report
Comments and Suggestions for Authors
The strength of the article is the issue of vaccination on influenza, especially if we refer to the pandemic population, so this type of research seeks to make prevention in vaccination every year of workers who are in contact with infected patients and avoid massive contagions in public health. The content of the article has an adequate structure, however it shows weaknesses in the theoretical instruction, it lacks to include public health research of empirical studies in health prevention institutions, to clarify the research design if it is experimental or not experimental, to define the type of study if it is descriptive or comparative and to characterize the sample of the population that received the vaccine, in the conclusions it is necessary to clarify if the objective of the research was achieved and if the public policy is to strengthen the prevention of diseases in the workers of the health institutions.
It is recommended that the observations be heeded, since this is a matter of public health interest to the population.
Comments on the Quality of English LanguageThe reading is clear and has good syntax in the structure and content of the text.
Author Response
Dear Reviewer,
please refer to the attached .doc file
We appreciated your valuable comments and we again thank you for the time spent in evaulating our work.
We improved the manuscript according to your suggestions.
Kind Regards,

Reviewer 2 Report
Comments and Suggestions for Authors
First of all, thank you for giving me the opportunity to evaluate the article. My suggestions about the article are below.
Abstract Section,
-The abstract sufficiently reflects the content of the study. It is written appropriately.
Introduction Section,
-The recommendation for vaccination of healthcare workers and the importance of disease transmission are given in detail. The gap in the literature is sufficiently stated.
-The study questions can be added at the end of this section.
Method Section,
-It would be appropriate to add which sampling method was used in the study and a priori sample size. How the proportions of healthcare workers in the sample were decided should be stated.
Inclusion and exclusion criteria are written and sufficient.
-Data collection tools should be detailed.
-Data analyses should be written appropriately, and it should be stated which software was used for data analyses, especially for AI analyses.
Results Section,
-Tables and graphs are prepared appropriately, and the findings reflect the content.
Discussion Section,
-Although the results are discussed by supporting the literature, the authors' comments on the results are lacking. The reasons for the similarities or differences of the results with the literature should be discussed in more detail. The limitations of the study are adequately written.
References Section,
-Appropriate and up-to-date.
Author Response
Dear Reviewer,
thank you again for the time spent in evaulating our work, we really appreciated your comments and amended the manuscript according to your suggestions.
Please refer to the .doc file for a point by point reply.
Kind Regards,

Reviewer 3 Report
Comments and Suggestions for Authors
Comment 1 - How can they show evidence of asking for informed consent?
Comment 2 - It would be important to clarify how the request for oral consent was made and then how the data was collected retrospectively.
Comment 3 - It would be important to standardise the presentation of the data in the results. Always with % or n and %.
Author Response
Dear Reviewer,
we thank you again for the time spent in evaulating our work, we really appreciated your comments and amended the manuscript according to your suggestions.
Please refer to the .doc file for a point by point reply.
Kind Regards,

Round 2
Reviewer 3 Report
Comments and Suggestions for Authors
Thank you for replying to my comments and for the improvements made to the work.
Author Response
Dear Reviewer,
here we upload the manuscript, which has been improved according to your comments.
Kind Regards,